# Cutaneous Nevoid Melanoma: A Retrospective Study on Clinico-Pathological Characteristics, with a Focus on Dermoscopic Features and Survival Analysis

**DOI:** 10.3390/cancers17010065

**Published:** 2024-12-29

**Authors:** Irene Russo, Emma Sartor, Rocco Cappellesso, Roberto Salmaso, Paolo Del Fiore, Gino Sartor, Antonella Vecchiato, Mauro Alaibac, Simone Mocellin

**Affiliations:** 1Soft-Tissue, Peritoneum and Melanoma Surgical Oncology Unit, Veneto Institute of Oncology IOV-IRCCS, 35128 Padua, Italy; irene.russo@iov.veneto.it (I.R.); antonella.vecchiato@iov.veneto.it (A.V.); simone.mocellin@unipd.it (S.M.); 2Dermatology Unit, Department of Medicine, University of Padova, 35128 Padua, Italy; emmasartor.93@gmail.com (E.S.); mauro.alaibac@unipd.it (M.A.); 3Pathological Anatomy Unit, Padua University-Hospital, 35128 Padua, Italy; rocco.cappellesso@gmail.com (R.C.); roberto.salmaso@aopd.veneto.it (R.S.); 4Primary Care Unit, 31011 Asolo, Italy; gino.sartor@yahoo.it; 5Department of Surgery, Oncology and Gastroenterology (DISCOG), University of Padua, 35128 Padua, Italy

**Keywords:** melanoma, nevoid melanoma, dermoscopic patterns, pathological features, dermoscopic features

## Abstract

Nevoid melanoma is a rare melanoma subtype that closely resembles a common nevus clinically and histologically. For this reason, diagnosis is easily missed. Both dermatologists and pathologists should be aware of this entity since its recognition might avoid severe consequences for the patient and medicolegal issues. Only a few papers on clinical and dermoscopic characteristics of nevoid melanoma are available. This study analyzes the clinical and pathological characteristics of nevoid melanoma in a population of patients affected by this rare subtype. It compares the prognosis and survival of these patients to data from classical melanoma featured in the literature. Additionally, by analyzing available dermoscopic images of nevoid melanoma, we aim to identify dermoscopic features that might help clinicians to suspect nevoid melanoma, reducing misdiagnosis.

## 1. Introduction

Nevoid melanoma (NeM) is a rare melanoma subtype that closely resembles a common nevus clinically and histologically. For this reason, diagnosis is easily missed [1,2,3]. In 1980, Levene first described a rare variant of melanoma and called it “verrucous and pseudo-naevoid melanoma” [4]. The term “nevoid malignant melanoma” was introduced in 1985 [5], concerning a series of malignant melanomas mimicking benign naevi at histological examination.

Clinical features of NeM are not distinctive [6]. It may present as a papule, nodule, or verrucous lesion with a variegated appearance; color ranges from pink/red to brown or black, and a mixed color pattern is frequent. NeM often mimics common popular naevi, but it sometimes resembles basal cell carcinomas or seborrheic keratosis [6]. Trunk and limbs are preferred locations for NeM [1,7,8,9,10].

Fast growth is one of the most useful clinical clues to suspect a NeM in a lesion that might otherwise present as a dermal nevus [11]. NeM is rarely reported in children [12].

Some authors distinguished three main clinical presentations of NeM, namely nevus-like NeM, amelanotic NeM, and multicomponent pattern NeM, sorted by frequency [7].

Overall, irregular dots/globules, multiple milia-like cysts, atypical vascular structures, blue-white veil, and irregular or eccentric hyperpigmented blotches are the most frequently detected dermoscopic features for NeM [7,8].

As for histology, nevoid melanoma mimics the architectural and cytological features of a common compound or intradermal melanocytic nevus at low power magnification. Classification of NeM as malignant is usually based on a combination of histologic clues and immunohistochemical findings. Indeed, at high power magnification variable nuclear atypia, irregular growth patterns with incomplete maturation, and/or dermal mitoses can be observed [6]. Ulceration might be an additional diagnostic feature. Ancillary techniques inconstantly aid the distinction between a mitotically active nevus and a nevoid melanoma [1,2,13]. A subset of NeM shows complete loss of p16 expression by immunohistochemistry. The value of PRAME in this setting is still unclear. Histological sub-categories of NeM (i.e., large nested NeM, papillomatous NeM, nodular NeM, or maturing NeM) have been identified, although they mainly meet a descriptive purpose [14]. BRAF and NRAS gene mutations are the most frequently detected mutations in NeMs [1].

Prognosis of the nevoid variant seems comparable to that of similarly staged conventional melanomas, with Breslow thickness confirmed as the most relevant prognostic factor, despite the bland histologic clues to malignancy [3,5,13,15]. Not rarely, stage IV disease uncovers an initial misdiagnosis of a nevoid melanoma, and final diagnosis of malignancy is made in a consultation practice. Thus, diagnostic error or delay may severely impact on overall prognosis in NeM [6].

Both dermatologists and pathologists should be aware of this entity, and clues to correct diagnosis might help them recognize this rare form of melanoma and avoid severe consequences for the patient and medicolegal issues.

The aim of this study is to analyze clinical and pathological characteristics of nevoid melanoma in a population of patients affected by this rare subtype and to compare prognosis and survival of these patients to data from classical melanoma featuring in the literature. Furthermore, by analyzing available dermoscopic images of nevoid melanoma, it makes an attempt to identify dermoscopic features that might help clinicians to suspect NeM, especially when mimicking common nevi.

## 2. Materials and Methods

This is a retrospective cohort study on nevoid melanoma in patients who were diagnosed and/or treated for primary melanoma at Veneto Institute of Oncology and at the University Hospital of Padua from August 1999.

### 2.1. Patients

All patients who were diagnosed and/or treated for NeM from 1999 at the Veneto Institute of Oncology (IOV) and at the University Hospital of Padua (Italy) were considered for inclusion in the study. Some patients of the cohort were diagnosed with NeM elsewhere in local level II centers or private clinics and then referred to IOV or to the University Hospital of Padua for histologic revision and successive treatment or for disease progression since the two participating hospitals are level III referral centers located in the Veneto Region in northeastern Italy; moreover, IOV is the referral center for melanoma in the Veneto region. Of note, some patients included in the study received a correct diagnosis after histological level III center revision of specimens misdiagnosed elsewhere, after the progression of the disease. Therefore, all patients included in the study received a histological diagnosis by expert dermatologists of either of the two level-III referral centers. The main inclusion criterion was a diagnosis of NeM based on to histologic criteria found in the literature; when the diagnosis of NeM was only probable, and differential diagnosis between other melanocytic lesions could not be excluded, patients were excluded.

### 2.2. Data Collection

Histological, clinical, and follow-up data were extracted from reports of scheduled visits, surgical intervention reports, and histologic examinations. Partial data were seldom available when the diagnosis of NeM dated back to pre-digital medical reporting in the two hospitals and/or when patients continued treatment for melanoma elsewhere. Tumors were classified at the time of diagnosis based on upon the eighth edition of the American Joint Committee on Cancer (AJCC) tumor [16], node, metastasis (TNM) staging system. Photo storage in a Dermox^®^ system for mole mapping was available for several patients in the database; a careful search for photos of melanocytic lesions lately diagnosed as NeM was conducted for these patients. Initially, we analyzed all the available pictures and classified the lesions into the following clinical categories: papule (elevated lesion up to 5 mm diameter), macule, nodule (elevated lesion more than 5 mm in diameter), and plaque. Secondly, we defined which dermoscopic patterns featured among those reported in the literature for classical melanoma and, specifically, for NeM. We considered: atypical pigmented reticulum; irregular globules and dots; asymmetric peripheric stare; eccentric pigmented blotches; blue-white veil; areas of regression with or without peppering; atypical vascular pattern with either linear irregular vessels or polymorphous vessels (any combination of different types of vascular structures including arborizing vessels, hairpin vessels, glomerular vessels, dotted vessels, linear irregular vessels, comma vessels, red lacunae) as typical dermoscopic patterns for classical melanoma. Patterns identified in the literature as more typical for NeM were irregular dots/globules, multiple milia-like cysts, and atypical vascular structures, classical melanoma-specific criteria with a multicomponent pattern, and hyperpigmented blotches [7,8,17]. Overall survival (OS) was calculated from the date of diagnosis to the date of death or last hospital access for any purpose. Disease(melanoma)-specific survival (DSS) was calculated from date of diagnosis of NeM to date of disease-related death. Disease-free survival (DFS) was calculated from the date of diagnosis to the date of recurrence, or the date of last visit/death for patients with no history of recurrence. Disease recurrence included local recurrence, regional lymph node metastases, regional skin/in-transit metastases and distant metastases. Local recurrence, regional lymph node metastases, regional skin/in-transit metastases were grouped during data analysis and considered together as locoregional recurrence, corresponding to a clinical and pathological stage III, as opposed to distant metastasis, corresponding to a stage IV. Patients with a diagnosis of one or more melanomas preceding NeM were excluded from follow-up and survival analysis since melanoma-related clinical events during their follow-up were not attributable to NeM. Data on new primary melanomas following NeM were also collected for our patients. However, new primary melanomas were not counted as clinical events in follow-up and did not impact prognosis as they were all diagnosed at an early stage during the scheduled melanoma follow-up visits.

### 2.3. Statistical Analysis

Clinical and histological data were summarized as mean, standard deviation, range for quantitative variables, and count and percentage for qualitative variables. Several missing values was reported. Clinical and histological characteristics in male and female patients were compared using the χ^2^ test for qualitative variables and the Student’s t-test for quantitative variables. Dermoscopic characteristics were summarized as several patients and as percentages of patients per characteristic and type of lesion. A survival analysis was conducted using the Kaplan-Meier method; results are presented as survival curves. Overall survival, melanoma-specific survival, and recurrence-free survival were calculated. A bivariate analysis was carried out to verify statistically significant differences in survival according to demographic, clinical, and histological variables. A *p*-value of less than 0.05 was considered statistically significant when calculated. Data were collected with Microsoft Excel and analyzed using R software version 3.4.1.

## 3. Results

### 3.1. Clinical and Histologic Characteristics

A total of 110 cases of nevoid melanoma were identified from 110 patients. Our sample showed a little male prevalence (*n* = 58, 52.7%). The age range was 15.2–88.3 years, with a mean of 53.05 years. Anatomic location was classified into four categories (head and neck; trunk; upper limbs; lower limbs), the most common being trunk, followed by lower limbs, upper limbs, and head and neck, respectively. Data on location were not available for six lesions. Of the patients in the study, 101 were from the Veneto region in northeastern Italy, seven patients were from other areas in Italy, and two were from other European countries. Eighty-seven (80.6%) resided in plain areas, 15 (13.9%) in coastal areas, five (4.6%) in the hills and one (0.9%) in the mountains. About histological features of the lesions, the mean Breslow thickness of NeM in our series was 1.4 (range 0.2 to 9 mm, SD 1.34); ulceration featured in 11 (10.2%) cases; lymphatic or neural invasion was found in four patients (3.7%); regression featured in 14 (12.8%); tumor-infiltrating lymphocytes were absent in 24 (24.5%) lesions, non-brisk in 66 (67.3%), brisk in eight (8.2%) and not defined in 12 cases; Four (4.0%) lesions had histological evidence of microsatellitosis; at least one mitosis per square millimeter was present in 83% of the tumors, and mitoses ranged from 0 to 16 per square millimeter, mean 2.34 (SD 2.97).

Regarding the TNM staging system, the T parameter for our NeMs featured as follows: T1a 39.4%; T1b 14.7%; T2 23.8%; T3 16.5%; T4 4.6%. One tumor was not radically excised, and histology confirmed deep margin involvement, therefore T parameter could not be established, this tumor was classified as Tx. None of the patients for whom histological reports of wide excision were available had evidence of residual melanoma in re-excision specimens.

The N parameter at diagnosis was N0 for 83.3% of the patients, and *N* > 1 for 16.7%. For 49 (49%) of the patients, sentinel lymph node biopsy (SLNB) was not conducted, because it was not indicated by protocols in T1a tumors, because the patient preferred not to undergo SNLB despite the protocols, or because straight lymph node dissection was recommended instead of SLNB based on clinical or imaging findings. Of note, in two cases, SLNB was not performed since a misdiagnosis of a benign lesion was made at first (see afterward).

SNLB was conducted in 51 (51%) patients; nodal metastases were present in 14/51 (14%). Lymph node dissection of regional nodes followed a positive SLNB in eight patients. Lymph node dissection was not preceded by SLNB and was directly performed in three patients in the series. Data on SLNB were unavailable for 10 patients since their staging data after diagnosis were missing. The M parameter at the time of diagnosis was M1 for three patients; M data at diagnosis were undetermined for 11 patients since full staging information was not available; interestingly, two of these 11 patients presented to IOV and received a NeM diagnosis at histological revision of histological specimens, after distant metastases occurred; therefore, N and M status had not been investigated at first, as the lesions had been misdiagnosed as a benign. Pathological TNM staging resulted in 66 (66.0%) stage I, 14 (14.0%) stage II, 17 (17%) stage III, and three (3%) stage IV classifications at the time of diagnosis. The clinical, histological, and staging features are summarized in Table 1.

### 3.2. Clinical and Dermoscopic Characteristics

Clinical pictures and dermoscopy of NeM were available for 24 patients. Based on the ABCDE criteria, the NeMs were divided into four groups according to the number of suspicion criteria for each lesion. A total of 37.5% (9/24) of the NeMs had 5/5 criteria, 8.3% (2/24) had 3/5 criteria, 16.7% (4/24) had 2/5 criteria, and finally, 37.5% (9/24) had 1/5 ABCDE criteria (Appendix A).

Evolving was present in 91.7% (22/24) of NeMs patients. Clinically, 11 (45.8%) lesions were papules, 9 (37.5%) were macules, two were nodules (8.3%), and two were plaques (8.3%). Therefore, 15 lesions were palpable and 9 were not palpable. All 24 NeM presented at least notes of brown color, 14 were medium brown, eight were light brown, two were dark brown; pink color featured in 11 lesions; red color featured in nine lesions, and in six cases, it was admixed with the pink color; blue color featured in three lesions, white color featured in two lesions, and black in another two lesions. Eight (33.33%) cases of NeM were monochromatic, nine (37.50%) revealed two colors, four (16.66%) cases revealed three colors, two (8.33%) cases revealed four colors, and one (4.1%) case revealed five different colors. Brown and pink/red were the most common colors in our clinically characterized NeMs. Overall features were typical of nevus-like NeM for 20 lesions and a multicomponent pattern (i.e., classical melanoma pattern) for four NeM lesions. No amelanotic NeM was featured among our cases. The main dermoscopic features of the 23 cases available are reported in Table 2, stratified according to the clinical appearance of the lesions (papule, nodule, macule, i.e., palpable lesions, and plaque, i.e., non-palpable lesions). No specific dermoscopic feature was identified for one lesion; therefore, it was classified as structureless. Figure 1, Figure 2, Figure 3 and Figure 4 show clinical and dermoscopic pictures of NeMs from our series.

### 3.3. Follow Up

Ten patients were excluded from follow-up analysis because they had a history of primary melanomas preceding NeM that could impact the prognosis, and clinical events in their follow-up could not be attributable to NeM. Multiple melanomas were the case in 15/110 total patients (13.63%), including five patients with a new melanoma after NeM and 10 patients with prior melanoma that were excluded from follow-up analysis. The follow-up period ranged from 0 to 7311 days, mean of 2042.89 days (SD 1757.80 days). Sixteen of the 100 patients considered for follow-up reported recurrence in their follow-up period. Local, in transit or regional lymph node recurrences were observed in a total of nine patients (some patients reporting both skin and lymph node involvement). However, only five of them had no subsequent evidence of distant metastatic disease, whereas four patients with loco-regional skin or lymph node recurrence progressed to distant metastatic disease. Distant metastatic disease as the first evidence of recurrence was the case in seven patients. Therefore, a total of 11/100 (11%) patients presented with or developed a metastatic disease from NeM during follow-up. At the end of the follow-up period, a total of 84 patients were alive, and 14 deceased. Ten died from melanoma, specifically NeM; two died from causes other than melanoma, and two died for unspecified causes, even though death from melanoma could not be excluded for the latter two patients. During follow-up, new primary melanoma following NeM was the case in 5/100 patients. However, all these melanomas were thinner than previous NeM lesions and did not require other treatment apart from wide excision and follow-up. Clinical events that occurred afterward during the follow-up period were all attributable to NeM and did not modify the NeM-related prognosis. Data on mutation analysis for NeM were available for five patients with an advanced stage III or IV classification at diagnosis or with disease recurrence. The mutation test was performed by either mass spectrometry techniques, Sanger DNA sequencing, or Real-Time PCR. Four patients had a V600E mutated BRAF gene, and one patient had a Q61K mutated NRAS gene. Data on follow-up are summarized in Appendix A.

### 3.4. Survival Analysis

In our series, five-years OS was 93.75% and 10-year OS was 79.37%. Five-years DSS for NeM was 93.77%, and 10-year DSS for NeM was 86.49%. Five-year DFS was 83.6%, and 10-year DFS 71.3% (Figure 5). In addition, OS, DSS for NeM and DFS according to the stage were comparable to those observed for melanoma in large population-based studies, especially as stage I, II, and III are concerned, despite the scant number of cases. (Figure 6, Figure 7 and Figure 8). On bivariate analysis, the only significant variables in determining prognosis for NeM were Breslow thickness (*p* = 0.0042), TNM stage at diagnosis, defined as I–II versus III–IV, (*p* = 0.00025), and clinical events during follow-up (*p*< 0.0001). Survival was not statistically different between male and female patients; different age categories, defined as <40 years and >40 years; residence in different geographic areas, Clark level of infiltration, defined as I–II level and III–IV level; and presence or absence of regression at histology (Figure 9).

## 4. Discussion

Diagnosis of certain melanocytic lesions still represents a challenge even for most experienced dermatologists and dermopathologists. However, in recent years some difficult-to-diagnose subtypes of melanoma, together with other melanocytic tumors, have gained importance, and there has been difficulty in defining diagnostic clues that may help clinicians to recognize those lesions and suspect their malignancy. An appropriate suspicion might prove crucial to alert pathologists and guide them to a correct diagnosis, since histologic features might also be misleading, resulting in potential pitfalls with terrible consequences for patients and leading to medicolegal issues. Since its first recognition as a distinct melanoma subtype by Levene in 1980, only case reports and a limited series of NeMs have been reported in the literature [4]. This is due both to the rarity of this entity and the initial uncertain collocation among melanocytic malignant lesions and blending with other entities. Initially, the literature focused on pathological difficulties in correctly diagnosing NeM [2,3,5,15,17,18,19]. However, it has recently become clear that clinical diagnosis was also devious and that it would be useful to warn clinicians about this rare entity and try and provide some clinical clues to diagnosis [7,9,20,21]. It should be stressed that not rarely, our pathologists come across these tumors in a consultation practice. Eight diagnoses of NeM were made after the revision of histological specimens in our series. Revision was required because the pathologist who first analyzed the lesion asked for a second opinion from to our referral center for its expertise in melanocytic lesions, or because the patients were referred to IOV after metastatic melanoma disease was discovered, despite an initial diagnosis of a benign nevus. In our series, diagnosis of NeM in consultation practice happened much less frequently than described by Zembowicz and Wong [3,15]. However, it has to be noted that a considerable share of our NeM cases occurred in patients included in a melanoma follow-up program, and only a smaller smaller number were diagnosed in routine melanoma prevention practice in the general population. In two women, the initial diagnosis was of benign nevus, and patients were referred to our center after metastatic spread occurred. Another lesion in a male patient was first diagnosed as a spitzoid neoplasm, but prompt revision revealed the correct diagnosis of NeM, and appropriate staging and management followed. No precise incidence of misdiagnosis in nevoid melanoma has been documented, although other authors describe its likelihood [2,3,6]. Unfortunately, we did not have the original histological diagnosis in our revised cases; therefore, we cannot provide reliable data on this incidence. Nevertheless, we feel that in some cases no final diagnosis was made before receiving a second opinion by our pathologists. To our knowledge, ours is currently the most numerous case series on NeM described in the literature. Most available papers focus on the histology of nevoid melanoma, and there are only a few case series on clinical aspects and with scant case records [8,17,20,21,22,23]. The clinical characteristics of our patients were substantially comparable to those featured in reviews on NeM. Mean age at diagnosis spanned from the fifth to sixth decade in life, with age ranging from adolescent/young adult age to old age. Only Idriss et al. reported a mean age of 61.88 years in their series of 43 patients [24]. No significant difference between males and females was noted in most papers [1,3,8,9,10]. The trunk was generally the main anatomical location, followed by limbs; there was no distinction between upper and lower limbs in most papers. Only one paper reported limbs as a prevalent location for NeM [10]. A slightly increased prevalence of trunk location in males relative to females, featured in the paper by Zembowicz et al., was confirmed in our patients [3]. This sex-dependent preference in trunk or limb location has been recognized also in classical melanoma [25]. Histological parameters varied among reports, even though there seemed to be a higher prevalence of early diagnosed NeM with a lower Breslow thickness among our patients if compared with many other reports. This might be reflected also in our clinical macroscopic and dermoscopic findings in the 23 analyzed lesions, as most of them were papules (*n* = 9) and macules (*n* = 9), and there were only two nodules and two plaques. Longo et al. in their series of 27 NeM cases, found a high prevalence of nodules and plaque-type lesions (n = 16 and n = 8, respectively), and only three papules; no macular lesions were described. The mean Breslow of 3.2 in their series also points at the severity of the lesions [7]. On the other hand, Pampena et al., in their review, included a macular lesion and an early diagnosed in situ NeM, presenting as a tiny maculo-papule in a patient followed with a scheduled skin examination for multiple melanomas [8,26]. This should raise suspicion among clinicians visiting patients with a prior history of melanoma, since small melanocytic lesions resembling banal dermal nevus should be carefully analyzed with dermoscopy, especially when recently arisen or new at mole mapping. Moreover, macular and papular lesions prevailed also in the paper by Bellinato et al. and mean Breslow thickness in their series was 0.6 mm, thus suggesting that macroscopic clinical presentation is also dependent on the timing of diagnosis and severity of the lesions [22]. Almost all patients that had pictures stored in the Dermox system in our series were already included in the IOV melanoma follow-up program; therefore, lesions could have been diagnosed at an earlier stage compared to lesions of patients referred to a level III level center after diagnosis was made. This could explain the high prevalence of macular and papular lesions and the few nodular or plaque-type lesions in the clinical presentation of the 24 cases with available pictures, as well as an average Breslow thickness that was lower in our patients if compared to other series [8,9] However, other studies presented an average Breslow thickness comparable to our findings, and even lower [1,10,22]. Diagnosing early NeM might be of major difficulty for clinicians visiting patients for the first time in routine practice elsewhere, with a possible relevant impact on prognosis. As for dermoscopic features, surprisingly, only nevus-like NeMs and multicomponent pattern NeMs featured among our cases, whereas there was no amelanotic NeM. All the lesions had at least some hue of brown color suggesting melanocytic origin [7]. Nevertheless, the high prevalence of pink and red color compared to the very low prevalence of blue, black and white color represents an important difference from classical melanoma. Dermoscopic patterns identified for NeM in the literature were confirmed also in our cases, with a high prevalence of atypical pigmented reticulum, irregular globules and dots, and hyperpigmented blotches. Possibly due to the absence of amelanotic lesions, the atypical vascular pattern and the milia-like cysts were less represented than in the series by Longo et al., whereas a blue-white veil and areas of regression were slightly more frequent. We confirmed the possibility that structureless NeM may be encountered in clinical practice [7]. Overall, a dermoscopy-specific diagnosis of NeM is still difficult, since high variability among in clinical presentations of NeM exists, and the scarcity of images available does not allow making solid conclusions but only provides some diagnostic clues. However, by excluding those lesions that closely mimic classical melanoma with a multicomponent pattern including atypical pigmentation, atypical distribution of globules and dots, and blue-white veil, we feel that special attention should be paid to atypical vascular patterns, multiple milia-like cysts, and a structureless pattern in lesions that may otherwise appear as banal dermal nevus to the naked eye; pink color admixed to red and brown color in irregular distribution are also clues. Evolution at the time of the lesion might be an important additional hint to raise suspicion and lead to correct management [22].

This study has some limitations. No histologic review was performed to reassess the diagnosis of nevoid melanoma. However, all diagnoses were made by pathologists with experience in melanocytic lesions and supervised by senior pathologists when doubts persisted after careful examination of the specimen. Our pathologists provided a second opinion also when the diagnosis of nevoid melanoma was made in minor centers and revised cases mistaken as a benign nevus. Histological sub-categories of NeM were not distinguished in our series; therefore, this aspect has not been investigated. Furthermore, clinical and dermoscopic pictures were taken with no standardization. Investigation for mutations was not conducted in every patient with metastatic disease, as major changes in clinical practice and the management of melanoma have occurred since the first registered case of NeM reported in our database. Moreover, management of melanoma in our patients was not homogeneous due to changes in clinical practice in recent decades. Therefore, no conclusions on therapy and management could be drawn from our study. Our data analysis provides a perspective on the subject. However, some missing data could have slightly altered the results. A prospective study on the subject could help to overcome retrospective study limitations about data availability and consistency in the future.

## 5. Conclusions

Based on our findings, we can conclude that main risk factors for severe disease in classical melanoma are confirmed for NeM. Prognosis is mainly dependent on Breslow thickness and on disease stage at diagnosis and deteriorates if recurrences is verified during follow-up for re-staging of the disease. Neither NeM does not have a better prognosis if compared with classical melanoma, nor it has a worse one. NeM’s malignant potential should not be underestimated, and the former definition of minimal deviation melanoma for this entity might have led to misleading conclusions about its prognosis. On the other hand, assumptions on the worst prognosis for NeM were made because its deceptive features often cause a delayed diagnosis that has serious consequences for the disease course [3]. The fact that BRAF and NRAS genes were mutated in the analyzed advanced-stage NeM patients, suggests that the biology of NeM corresponds to that of classical melanoma, further supporting the interpretation of this melanoma as a diagnostic challenge, rather than a distinct biological entity. Moreover, the high prevalence in our series of patients with one or more classical melanomas in addition to NeM, also strengthens the belief that NeM should be assimilated into classical melanoma regarding its biology and consequent clinical behavior [3,6].

At present, clinical and histological recognition remains the preeminent issue for this type of melanoma. Overall, a dermoscopy- specific diagnosis of NeM is still difficult, since high variability in the clinical presentation of NeM exists. However, some dermoscopic patterns and the presence of evolution of a melanocytic lesion could suggest NeM. By excluding those lesions presenting with a multicomponent pattern closely mimicking classical melanoma, we feel that special attention should be paid to the atypical vascular pattern, multiple milia-like cysts, structureless pattern in lesions that may otherwise appear as banal dermal nevi to the naked eye; pink color admixed to red and brown color in irregular distribution are also clues. Multicenter research aiming to better characterize clinical and dermoscopic features of NeM should be encouraged because clinical suspicion might prove crucial to further pathological analysis and recognition of this rare melanoma subtype.

## Figures and Tables

**Figure 1 cancers-17-00065-f001:**
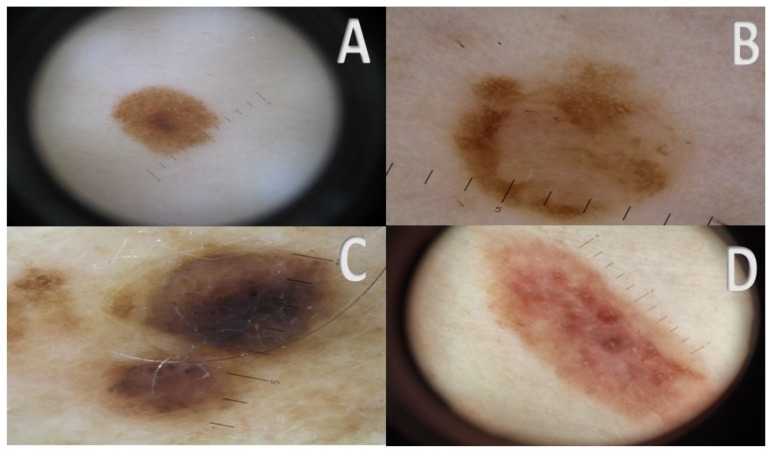
(**A**–**D**) Clinical and dermoscopic presentation of different macroscopic presentations of NeM. (**A**) macular NeM. (**B**) papular NeM. (**C**) nodular NeM. (**D**) plaque-type NeM. A prevalence of brown color in various hues characterizes these lesions. Pink color mixes with brown color in one of them (**D**).

**Figure 2 cancers-17-00065-f002:**
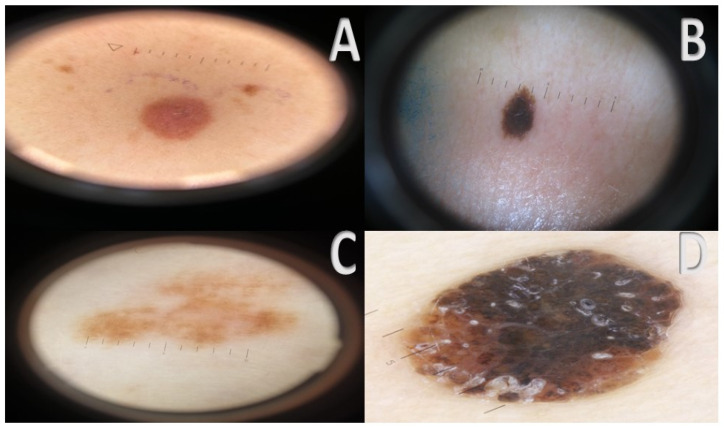
(**A**–**D**) Naevus-like nevoid melanoma. All lesions resemble common naevi, despite heterogeneous clinical and dermoscopic presentation. At dermoscopy, very dark to light brown color (**A**–**D**) and pink color (**A**) prevail, and features vary from structureless lesions (**A**), to hyperpigmented lesions (**B**) with nevus-like (**B**) or verrucous nevus like (**D**) appearance, and irregular pigmented reticulum (**C**). Notice the periferic stare at dermoscopy in (**D**).

**Figure 3 cancers-17-00065-f003:**
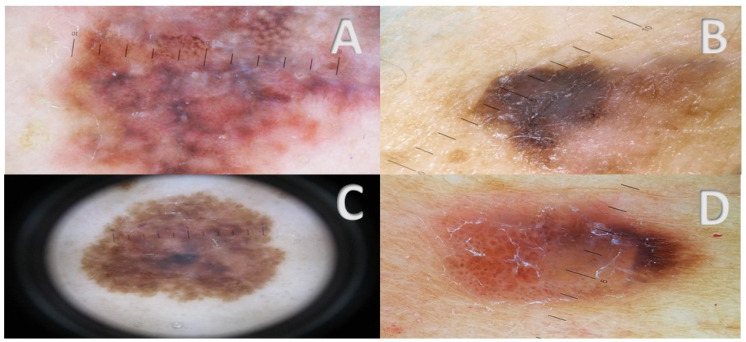
(**A**–**D**) NeM reveals a classical melanoma pattern. Melanoma-specific criteria featuring in these tumors are atypical pigmented reticulum (**A**,**C**), eccentric pigmented blotches (**B**,**D**), blue-white veil (**A**–**C**), areas of regression (**B**), atypical vascular pattern with glomerular vessels (**D**).

**Figure 4 cancers-17-00065-f004:**
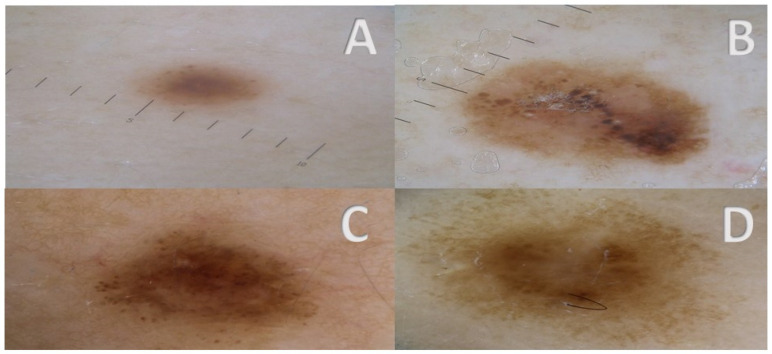
(**A**–**D**) Other NeM lesions displaying features identified as more frequently associated with NeM, i.e., irregular dots/globules (**A**–**D**), atypical pigmented reticulum (**D**), hyperpigmented blotches (**C**).

**Figure 5 cancers-17-00065-f005:**
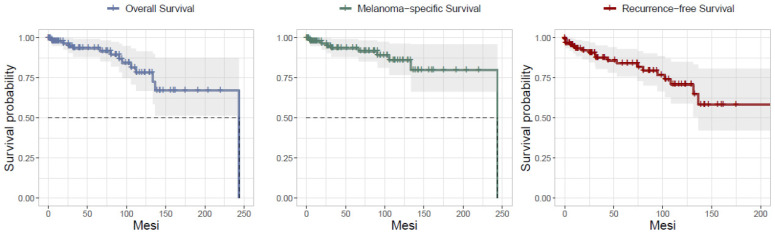
Overall Survival, Disease (Melanoma) Specific Survival, Disease Free Survival; *n* = 100.

**Figure 6 cancers-17-00065-f006:**
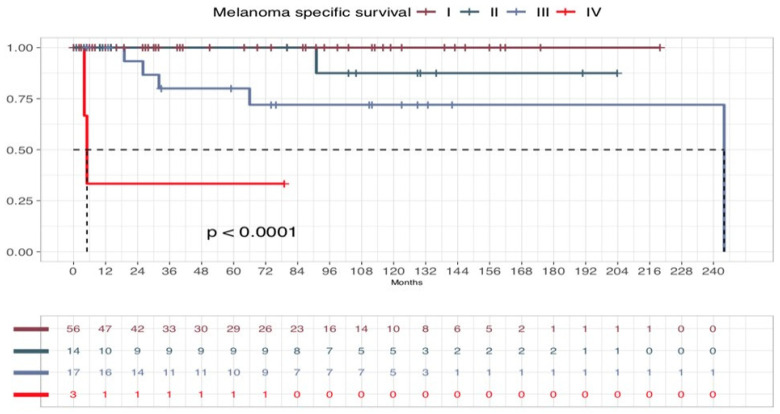
Melanoma-Specific Survival (MSS) curve according to the stage in patients with stage I to IV melanoma.

**Figure 7 cancers-17-00065-f007:**
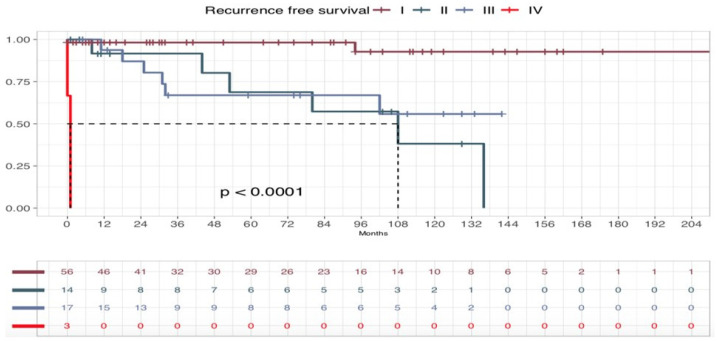
Recurrence-Free Survival (RFS) curve according to the stage in patients with stage I to IV melanoma.

**Figure 8 cancers-17-00065-f008:**
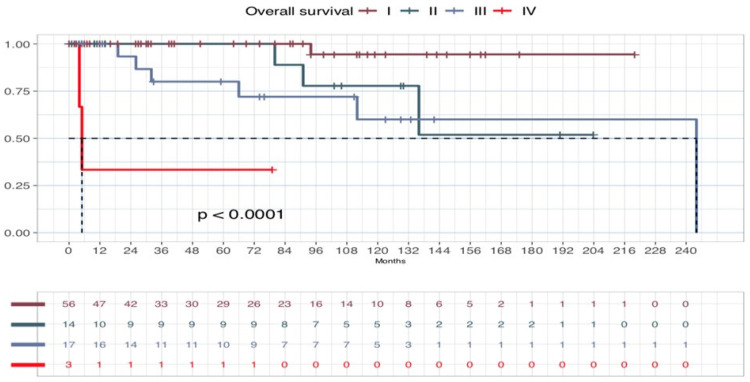
Overall Survival (OS) curve according to the stage in patients with stage I to IV melanoma.

**Figure 9 cancers-17-00065-f009:**
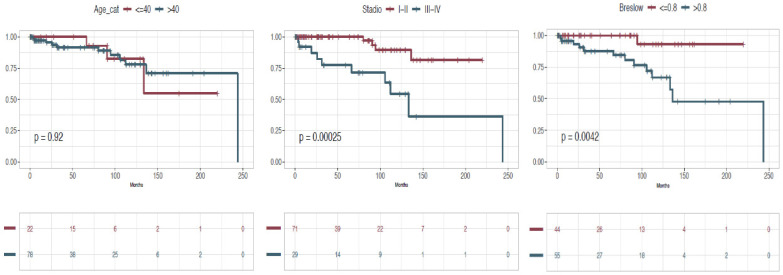
Kaplan-Meier plots for nevoid melanoma.

**Table 1 cancers-17-00065-t001:** Clinical, histological and staging features of NeM at diagnosis.

	Female (*N* = 52)	Male (*N* = 58)	Total (*N* = 110)	*p* Value
Age				0.027 ^1^
Mean (SD)	49.992 (15.290)	55.789 (17.075)	53.049 (16.441)	
Range	25.153–86.146	15.190–88.307	15.190–88.307	
Anatomic location				0.571 ^2^
head and neck	5 (10.4%)	8 (14.3%)	13 (12.5%)	
lower limb	14 (29.2%)	10 (17.9%)	24 (23.1%)	
trunk	22 (45.8%)	28 (50.0%)	50 (48.1%)	
upper limb	7 (14.6%)	10 (17.9%)	17 (16.3%)	
Breslow thickness				0.825 ^1^
N-Miss	0	1	1	
Mean (SD)	1.244 (1.024)	1.543 (1.570)	1.400 (1.340)	
Range	0.200–6.240	0.200–9.000	0.200–9.000	
Lymphovascular invasion				0.031 ^2^
absent	47 (92.2%)	57 (100.0%)	104 (96.3%)	
present	4 (7.8%)	0 (0.0%)	4 (3.7%)	
**Regression**				0.697 ^2^
absent	46 (88.5%)	49 (86.0%)	95 (87.2%)	
present	6 (11.5%)	8 (14.0%)	14 (12.8%)	
**Clark level**				0.997 ^2^
II	2 (4.3%)	2 (3.7%)	4 (4.0%)	
III	15 (31.9%)	18 (33.3%)	33 (32.7%)	
IV	29 (61.7%)	33 (61.1%)	62 (61.4%)	
V	1 (2.1%)	1 (1.9%)	2 (2.0%)	
**TIL**				0.202 ^2^
absent	14 (31.8%)	10 (18.5%)	24 (24.5%)	
present, brisk	2 (4.5%)	6 (11.1%)	8 (8.2%)	
present, non-brisk	28 (63.6%)	38 (70.4%)	66 (67.3%)	
**Ulceration**				0.654 ^2^
absent	46 (88.5%)	51 (91.1%)	97 (89.8%)	
present	6 (11.5%)	5 (8.9%)	11 (10.2%)	
**Mitoses per mm^2^**				0.368 ^1^
N-Miss	3	1	4	
Mean (SD)	2.776 (3.630)	1.982 (2.232)	2.349 (2.973)	
Range	0.000–16.000	0.000–10.000	0.000–16.000	
**Microsatellites**				0.258 ^2^
absent	49 (94.2%)	57 (98.3%)	106 (96.4%)	
present	3 (5.8%)	1 (1.7%)	4 (3.6%)	
**pTNM stage 8th ed.**				0.901 ^2^
I	30 (63.8%)	36 (67.9%)	66 (66.0%)	
II	7 (14.9%)	7 (13.2%)	14 (14.0%)	
III	9 (19.1%)	8 (15.1%)	17 (17.0%)	
IV	1 (2.1%)	2 (3.8%)	3 (3.0%)	
**SLNB_Status**				0.228 ^2^
not conducted	25 (54.3%)	24 (44.4%)	49 (49.0%)	
conducted, negative	13 (28.3%)	24 (44.4%)	37 (37.0%)	
conducted, positive	8 (17.4%)	6 (11.1%)	14 (14.0%)	

^1^ quantitative variables, ^2^ qualitative variables.

**Table 2 cancers-17-00065-t002:** Dermoscopic features of NeM stratified by clinical presentation.

	Non-Palpable Lesion	Palpable Lesion			
	Macule 37.5% (*N* = 9)	Nodule 8.3% (*N* = 2)	Papule 45.8%(*N* = 11)	Plaque 8.3% (*N* = 2)	Total (*N* = 24)
atypical pigmented reticulum	7 (46.7%)	1 (6.7%)	5 (33.3%)	2 (13.3%)	15
irregular globules and dots	2 (16.7%)	1 (8.3%)	7 (58.3%)	2 (16.7%)	12
asymmetric periferic strae	0 (0.0%)	0 (0.0%)	2 (100.0%)	0 (0.0%)	2
blue-white veil	2 (50.0%)	1 (25.0%)	1 (25.0%)	0 (0.0%)	4
areas of regression	1 (50.0%)	1 (50.0%)	0 (0.0%)	0 (0.0%)	2
atypical vascular pattern	0 (0.0%)	0 (0.0%)	2 (66.7%)	1 (33.3%)	3
multiple milia-like cysts	1 (20.0%)	1 (20.0%)	2 (40.0%)	1 (20.0%)	5
multicomponent pattern	1 (20.0%)	0 (0.0%)	0 (0.0%)	0 (0.0%)	1
hyperpigmented blotches	4 (57.1%)	1 (14.3%)	1 (14.3%)	1 (14.3%)	7
structureless	1 (100.0%)	0 (0.0%)	0 (0.0%)	0 (0.0%)	1
brown color					
- medium brown	7 (50.0%)	2 (14.3%)	4 (28.6%)	1 (7.1%)	14
- light brown	2 (25.0%)	0 (0.0%)	5 (62.5%)	1 (12.5%)	8
- dark brown	0 (0.0%)	0 (0.0%)	2 (100.0%)	0 (0.0%)	2
pink color	4 (36.4%)	1 (9.1%)	4 (36.4%)	2 (18.2%)	11
red color	2 (22.2%)	1 (11.1%)	4 (44.4%)	2 (22.2%)	9
color-blue	2 (66.7%)	1 (33.3%)	0 (0.0%)	0 (0.0%)	3
white color	0 (0.0%)	1 (50.0%)	0 (0.0%)	1 (50.0%)	2
black color	2 (100.0%)	0 (0.0%)	0 (0.0%)	0 (0.0%)	2

## Data Availability

The datasets presented in this study can be found in online repository. The names of the repository and accession number(s) can be found here: https://doi.org/10.5281/zenodo.14563905 (accessed on 27 December 2024).

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
