# Peer review of "Cutaneous Nevoid Melanoma: A Retrospective Study on Clinico-Pathological Characteristics, with a Focus on Dermoscopic Features and Survival Analysis"

_cancers, 2024, doi:10.3390/cancers17010065_

Round 1
Reviewer 1 Report
Comments and Suggestions for Authors
The article is very good. The methodology used is robust, the number of patients is high (110 cases), finally the results are well explained.
Nevoid Melanoma represents a diagnostic challenge for both the dermatologist and the pathologist, and misdiagnosis is possible.
The strength of the study is the presence of dermatoscopic evaluation of 23 lesions: a good number considering the "rarity" of this type of melanoma. Data collection was performed starting from 1999 allowing an appropriate follow up.
I point out two typos in the acronym NeM on lines 76 and 78, which I recommend to correct.
Author Response
Thank you very much for your comment.
The acronym NeM has been corrected in the manuscript. Please see lines 76-78
Reviewer 2 Report
Comments and Suggestions for Authors
The authors investigated clinicopathological characteristics of nevoid melanoma retrospectively. I have some comments and questions below.
Major points
1. To identify clinical and dermoscopic features of nevoid melanoma, the authors should clearly describe the definition of nevoid melanoma in this study. Generally, a number of melanoma cases resemble melanocytic nevus and these two diseases are differentiated carefully. I am unsure by what diagnosis criteria the authors select nevoid melanoma cases. In Figures 1 to 4, some cases resemble solar lentigo rather than melanocytic nevus.
2. Figures 6 to 8 should be presented appropriately. A part of the graphs is cut off. Full of the graphs should be presented.
3. Clinical manifestations such as asymmetry, border irregularity, color variegation, diameter of the tumor, and elevation should be evaluated in nevoid melanoma cases according to ABCD rules of melanoma.
Minor point
In line 64, “three mail clinical presentation” should be corrected to “three main clinical presentation.”
In line 76, “Nem” should be “NeM.”
In lines 175 to 176, The sentence “Most patients were 175 males (n = 58, 52.7%)” should be modified. A number of NeM patients are female in this study.
Author Response
Thank you for your constructive comment.
We revised the manuscript according to your suggestions.
Major points
1) Thank you for this important comment. Nevoid melanoma cases were selected on the basis of histologic diagnosis. Once the cases were selected, we analyzed the macroscopic and dermoscopic images in order to identify the characteristic features of these lesions. This is clearly specified in the manuscript. Please see lines 112-116
I agree that some lesions did not have clinical features suggestive of melanoma, which was later diagnosed histologically. Patients referred to our center are high-risk patients for melanoma who undergo periodic digital videodermatoscopy and radical surgical excision of melanocytic lesions that show characteristics of atypia and/or are evolving.
2) Figures 5 to 7 are cut off in the manuscript version but are fully and clearly visible in the figures and tables file.
3) Thank you for your valuable comment. We had actually given little space in the manuscript to describing the macroscopic clinical features of the lesions compared to the dermoscopic ones.
According to your suggestion, we have included in the manuscript a supplementary table and description regarding the ABCDE features of the 24 analyzed lesions.
Please see Supplementary Table.2 (ABCDE features) and lines 219-224 in the manuscript.
Minor points
The sentence “three mail clinical presentation” has been corrected in “three main clinical presentation”. Please see line 64
The acronym “Nem” has been corrected in “NeM”. Please see lines 76-78
The sentence “Most patients were males (n=58, 52.7%)” has been modified in “Our sample showed a little male prevalence (n=58, 52.7%)”. Please see lines 175-176
Round 2
Reviewer 2 Report
Comments and Suggestions for Authors
The authors responded to all comments and questions adequately. I don't have any more comments.